# Remote early detection of SARS-CoV-2 infections using a wearable-based algorithm: Results from the COVID-RED study, a prospective randomised single-blinded crossover trial

Laura C. Zwiers[1,2*☯], Timo B. Brakenhoff[1☯], Brianna M. Goodale[1,3], Duco Veen[4,5], George S. Downward[2,6], Vladimir Kovacevic[3,7], Andjela Markovic[3,8], Marianna Mitratza[2], Marcel van Willigen[1], Billy Franks[1], Janneke van de Wijgert[9], Santiago Montes[10], Serkan Korkmaz[11], Jakob Kjellberg[11], Lorenz Risch[12,13,14], David Conen[15], Martin Risch[12,16], Kirsten Grossman[12,13], Ornella C. Weideli[12,13], Theo Rispens[17,18], Jon Bouwman[1], Amos A. Folarin[19,20,21], Xi Bai[19], Richard Dobson[19], Maureen Cronin[3‡], Diederick E. Grobbee[1,2‡], On behalf of the COVID-RED consortium[¶]

1 Julius Clinical, Zeist, The Netherlands, 2 Department of Global Health and Bioethics, Julius Center for Health Sciences and Primary Care, University Medical Center Utrecht, Utrecht, The Netherlands, 3 Ava AG, Zürich, Switzerland, 4 Department of Methodology and Statistics, Utrecht University, Utrecht, The Netherlands, 5 Optentia Research Programme, North-West University, Potchefstroom, South Africa, 6 Department of Environmental Epidemiology, Institute for Risk Assessment Sciences (IRAS), Utrecht University, Utrecht, The Netherlands, 7 The Institute for Artificial Intelligence Research and Development of Serbia, Novi Sad, Serbia, 8 Department of Psychology, University of Fribourg, Fribourg, Switzerland, 9 Department of Epidemiology and Health Economics, Julius Center for Health Sciences and Primary Care, University Medical Center Utrecht, Utrecht, The Netherlands, 10 Roche Diagnostics Nederland B.V., Almere, The Netherlands, 11 VIVE, Copenhagen, Denmark, 12 Laboratory Dr. Risch, Vaduz, Liechtenstein, 13 Faculty of Medical Sciences, Insitute of Laboratory Medicine (ILM), Private University in the Principality of Liechtenstein (UFL), Triesen, Principality of Liechtenstein, 14 Center of Laboratory Medicine, University Institute of Clinical Chemistry, University of Bern, Bern, Switzerland, 15 Population Health Research Institute, McMaster University, Hamilton, Canada, 16 Central Laboratory, Kantonsspital Graubünden, Chur, Switzerland, 17 Sanquin Research and Landsteiner Laboratory, Amsterdam UMC, Amsterdam, The Netherlands, 18 Amsterdam Institute for Infection and Immunity, Amsterdam, The Netherlands, 19 Institute of Health Informatics, University College London, London, United Kingdom, 20 NIHR Biomedical Research Centre, University College London Hospitals NHS Foundation Trust, London, United Kingdom, 21 Department of Biostatistics and Health Informatics, Institute of Psychiatry, Psychology and Neuroscience, King's College London, London, United Kingdom

¶Membership of the COVID-RED consortium is provided in the Acknowledgements
☯ These authors contributed equally to this work.
‡ MC and DEG also contributed equally to this work.
* laura.zwiers@juliusclinical.com

## Abstract

### Background

Rapid and early detection of SARS-CoV-2 infections, especially during the pre- or asymptomatic phase, could aid in reducing virus spread. Physiological parameters measured by wearable devices can be efficiently analysed to provide early detection of infections. The COVID-19 Remote Early Detection (COVID-RED) trial investigated the use of a wearable device (Ava bracelet) for improved early detection of SARS-CoV-2 infections in real-time.

---

**Data availability statement:** All relevant, anonymised, data from the COVID-RED project was stored and made publicly available through DataverseNL. Prior to their publication, the data were processed and separated into relevant domains, allowing external researchers to efficiently utilise the data while protecting subjects' privacy. Available from: https://dataverse.nl/dataset.xhtml?persistentId=doi:10.34894/FW9PO7

**Funding:** The COVID-RED project has received funding from the Innovative Medicines Initiative (https://www.imi.europa.eu) 2 Joint Undertaking under grant agreement No 101005177. This Joint Undertaking receives support from the European Union's Horizon 2020 (https://ec.europa.eu/programmes/horizon2020/) research and innovation programme and EFPIA (https://www.efpia.eu/). Disclaimer: The research leading to these results was conducted as part of the COVID-RED Consortium. This paper only reflects the personal views of the stated authors. The funding body has no role in the design of the study; the collection, analysis, and interpretation of the data; and the writing of the manuscript.

**Competing interests:** The authors have read the journal's policy and have the following competing interests: Laura Zwiers, Marcel van Willigen, Jon Bouwman and Diederick Grobbee are current employees of Julius Clinical BV. Timo Brakenhoff, Brianna Goodale and Duco Veen are former employees of Julius Clinical BV. Billy Franks is a former employee of Julius Clinical BV and now an employee of Haleon. Brianna Goodale, Vladimir Kovacevic, Andjela Markovic and Maureen Cronin are past employees of Ava AG. Marianna Mitratza is a current employee of P95 CVBA. Lorenz Risch and Martin Risch are current employees and key shareholders of Dr Risch Medical Laboratory. Kirsten Grossman and Ornella Weideli are current or former employees of Dr Risch Medical Laboratory. David Conen has received consulting fees from Roche Diagnostics, outside of the current work. There are no patents, products in development or marketed products associated with this research to declare. These competing interests do not alter our adherence to PLOS ONE policies on sharing data and materials.

## Trial design

Prospective, single-blinded, two-period, two-sequence, randomised controlled cross-over trial.

## Methods

Subjects wore a medical device and synced it with a mobile application in which they also reported symptoms. Subjects in the experimental condition received real-time infection indications based on an algorithm using both wearable device and self-reported symptom data, while subjects in the control arm received indications based on daily symptom-reporting only. Subjects were asked to get tested for SARS-CoV-2 when receiving an app-generated alert, and additionally underwent periodic SARS-CoV-2 serology testing. The overall and early detection performance of both algorithms was evaluated and compared using metrics such as sensitivity and specificity.

## Results

A total of 17,825 subjects were randomised within the study. Subjects in the experimental condition received an alert significantly earlier than those in the control condition (median of 0 versus 7 days before a positive SARS-CoV-2 test). The experimental algorithm achieved high sensitivity (93.8–99.2%) but low specificity (0.8–4.2%) when detecting infections during a specified period, while the control algorithm achieved more moderate sensitivity (43.3–46.4%) and specificity (66.4–65.0%). When detecting infection on a given day, the experimental algorithm also achieved higher sensitivity compared to the control algorithm (45–52% versus 28–33%), but much lower specificity (38–50% versus 93–97%).

## Conclusions

Our findings highlight the potential role of wearable devices in early detection of SARS-CoV-2. The experimental algorithm overestimated infections, but future iterations could finetune the algorithm to improve specificity and enable it to differentiate between respiratory illnesses.

## Trial registration

Netherlands Trial Register number NL9320.

## Introduction

The severe acute respiratory syndrome coronavirus 2 (SARS-CoV-2), associated with the coronavirus disease 2019 (COVID-19), caused a global pandemic leading to over 775 million cases and seven million deaths worldwide [1]. During the pandemic, the standard approach to controlling the spread of SARS-CoV-2 relied upon individuals seeking a diagnostic test when developing symptoms and isolating after being

exposed. However, this approach was complicated by the fact that most infected individuals became infectious before symptom onset, with an average incubation period of 6.57 days [2]. During this incubation period, the viral load of SARS-CoV-2 increases, such that pre-symptomatic individuals can transmit the virus unknowingly. It has been suggested that over 40% of infected individuals were asymptomatic [3], and that pre- and asymptomatic cases were responsible for more than half of all COVID-19 transmissions [4–6].

Rapid and early detection of SARS-CoV-2 during the pre- or asymptomatic phase could facilitate isolation of cases before transmissions occur. Inviting individuals exposed to an infected person for testing (as was a common temporary policy in many countries and was recommended by the World Health Organisation [7,8]) ignores individuals unaware of an exposure. Frequent testing of healthy populations poses logistical and budgetary challenges, while screening for easy-to-measure physiological signs that predict infection prior to symptom onset could facilitate timely identification of infected individuals while limiting the operational and financial impact [9–11].

Physiological monitors that can detect increased body temperature and pulse rate related to fever [12], which is one of the most common symptoms of COVID-19 [13,14], are commercially available, including wearable devices. Many wearable devices also register changes in breathing rates, associated with shortness of breath and tachycardia [13,14]. Ingested by machine learning algorithms, these physiological signals can be efficiently processed and analysed to support early detection of SARS-CoV-2 infections.

A systematic review published in 2022 identified multiple studies that support the use of wearable devices to detect SARS-CoV-2 infection prior to symptom onset [9]. However, these studies were relatively small and used retrospective study designs which increased their potential for bias while limiting their ability to evaluate efficacy in a real-world context. After publication of the review, a prospective study of 1,163 individuals in Liechtenstein [10] reported that the use of a wearable device (the Ava bracelet) could detect SARS-CoV-2 infections two days prior to symptom onset in 68% of cases. The study was performed in a sample of relatively young individuals (mean age of 44, maximum age of 51) and therefore lacked generalisability to older and more vulnerable populations. A prospective study in 2021 aimed to detect early infection in 3,318 participants using data from various wearable devices [15]. This study detected most pre- and asymptomatic individuals, with presymptomatic individuals identified at a median of three days before symptom onset. However, many asymptomatic cases were likely missed due to reliance on self-reported positive tests in this study. Enrolling 38,911 individuals between March 2020 and April 2021, another prospective study used self-reported symptoms as well as wearable device data for SARS-CoV-2 detection [16]. While high performance metrics (Area Under the Curve [AUC]) for both symptomatic (AUC = 0.83) and asymptomatic (AUC = 0.74) cases were achieved, performance in presymptomatic cases was not reported on. A prospective study in the United States in 2020 achieved similar performance when differentiating between cases and non-cases among symptomatic individuals (AUC = 0.80) [17] but did not investigate performance for asymptomatic infections. Additionally, none of these prospective studies included a control group.

The COVID-19 Rapid Early Detection (COVID-RED) study was organised in May 2020 by a consortium of academic and industry partners to investigate the possibility of using physiological data from a wearable medical device for improved early detection of SARS-CoV-2. This trial included nearly 18,000 individuals living in the Netherlands, thereby comprising one of the largest randomised trials examining early detection of SARS-CoV-2 in real-time to date. Subjects wore a medical device measuring various physiological parameters on their wrist while sleeping. Using algorithms based on physiological parameters, as well as subjects' self-reported symptoms, this study aimed to improve the detection of SARS-CoV-2 and, in particular, pre- or asymptomatic infections.

Using laboratory-confirmed SARS-CoV-2 infections as the gold standard, this study aimed to compare the performance of two algorithms in their ability to detect first-time SARS-CoV-2 infection, including early detection of pre- or asymptomatic cases: (1) an algorithm ingesting data from a wearable medical device coupled with self-reported daily symptom data (i.e., experimental condition), and (2) an algorithm using self-reported daily symptom data only (i.e., control condition).

 

## Materials and methods

### Study design

COVID-RED was a single-blinded, two-period, two-sequence, randomised controlled crossover trial. The study was reviewed and approved by the ethical review committee at the University Medical Centre Utrecht and registered in the Netherlands Trial Register on February 18, 2021, with number NL9320. The study protocol has been previously published [18].

### Subjects

Eligible subjects recruited from various sources including public outreach campaigns and pre-existing cohort studies were enrolled during the first half of 2021. All subjects were Dutch speaking residents of the Netherlands over the age of 18 who had not knowingly had a prior SARS-CoV-2 infection and were willing to use a wearable device alongside an accompanying smartphone application. Exclusion criteria were prior self-reported SARS-CoV-2 infection, participation in another COVID-19 clinical trial, the use of an electronic implanted device, pregnancy, or suffering from cholinergic urticaria (a known contraindication for the wearable device). While initially an exclusion criterion in the first month of subject recruitment, subjects who received a COVID-19 vaccine were enrolled when it became clear that rapid uptake of vaccinations in the Netherlands was inevitable.

As the severity and predispositions to SARS-CoV-2 infection can vary based on demographic and health features [19], both "normal" and "high" risk individuals were actively recruited. High-risk individuals were defined as individuals fulfilling any of the following self-reported criteria: age of 70 years or older; body mass index (BMI) over 40; employed in a hospital or care home with regular patient/client contact; having a chronic medical condition; or, long term use of specific medications or treatments (e.g., medication for high blood pressure, heart disorders, diabetes, human immunodeficiency virus, chemotherapy, immunotherapy, radiotherapy, immunosuppressive medication).

All subjects gave informed consent prior to enrolling in the study and could withdraw from the study at any time for any reason.

### Randomisation and masking

Recruited subjects received a wearable device (the Ava bracelet) and were asked to download a smartphone application on their personal device ("Ava COVID-RED"). Subjects were randomised 1:1 to either Sequence 1 (experimental condition followed by control condition), or Sequence 2 (vice versa). The study started with a learning phase (maximum three months) for determining baseline physiological parameters, followed by three months in period 1 (in which the first condition was applied), then three months in period 2 (in which the second condition was applied). Subjects were blinded to condition at all times by wearing the Ava bracelet and having access to their data, even if the wearable-generated data was not ingested by the algorithm.

### Wearable device and symptom diary

The Ava bracelet (Ava AG, Zurich, Switzerland) was an FDA-cleared and CE-certified fertility aid that was worn on the user's wrist while sleeping. The Ava bracelet contains three sensors that measure five physiological parameters every 10 seconds: respiratory rate, heart rate, heart rate variability (in milliseconds), wrist-skin temperature (in degrees Celsius), and skin perfusion.

All subjects wore the Ava bracelet while sleeping and synchronised it with the "Ava COVID-RED" smartphone application upon waking. In the app's daily diary, subjects were instructed to record any physical symptoms they experienced (e.g., headache, nausea), as well as factors that could affect their physiological parameters (e.g., alcohol, drug, or medication use), and diagnostic SARS-CoV-2 test results when they had undergone testing. Subject compliance with bracelet

and app usage was periodically reviewed by the study team, who contacted subjects with low compliance for additional follow-up. Help desks were set up to allow subjects to report any technical issues and adverse events experienced during the trial (e.g., rash from the wristband).

For both the experimental and control conditions, algorithms were applied to predict the presence of a SARS-CoV-2 infection. The control condition's algorithm was designed to mimic the Dutch SARS-CoV-2 testing policy; only subjects reporting certain symptoms (e.g., common cold symptoms, coughing, shortness of breath, elevated temperature or fever, and sudden loss of smell and/or taste) were advised to get tested. For the experimental condition, a machine learning algorithm was developed that ingested app-reported data, as well as the physiological parameters measured by the Ava bracelet. Both algorithms could trigger a "red alert", which indicated that a subject should seek SARS-CoV-2 testing as data suggested a potential infection. Fig 1 shows the messages that subjects could receive from the app.

For the algorithm to be able to use the physiological data, a baseline of subjects' "normal" physiological patterns was needed. This baseline was determined during the learning phase, and a first version of the algorithm was applied in real-time during period 1. All main analyses, however, were performed based on the predictions from version 2 of the algorithm, which was developed using data from both the learning phase and period 1 and implemented in real-time during period 2. Upon study completion, version 3 of the algorithm was developed using all available data. This algorithm was applied retrospectively only and is beyond the scope of the current paper. Fig 2 shows a schematic overview of the different study periods and algorithms.

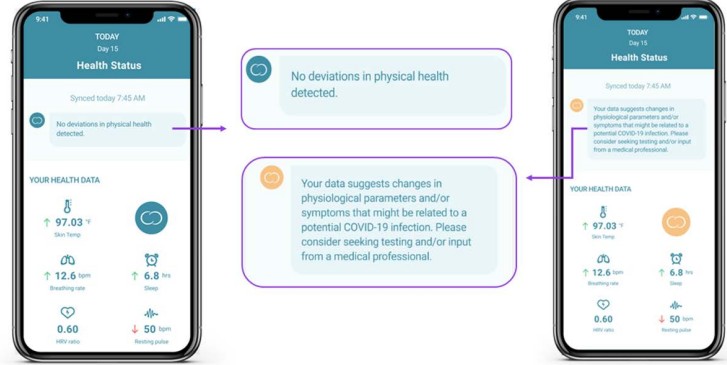

**Fig 1. Illustration of the in-app messages given in case of unlikely indication for infection (left) and in case of a "red alert" (right).**

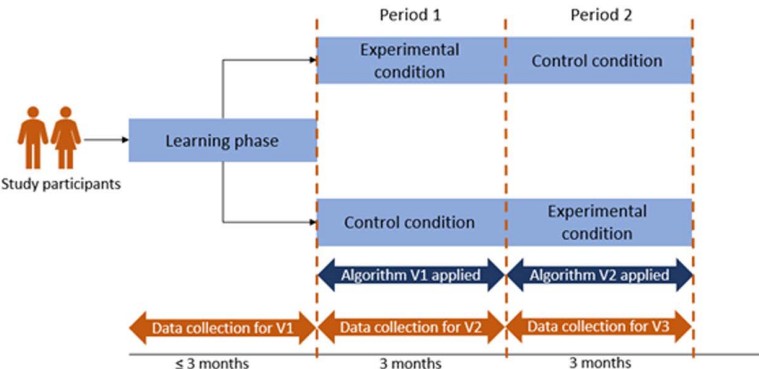

**Fig 2. Schematic illustration of the study periods and algorithms applied during the COVID-RED study.**

## SARS-CoV-2 testing

When subjects received a "red alert", they were advised to get tested by PCR and/or antigen test. Subjects were asked to seek testing at the Dutch Public Health Service. When this was not possible (e.g., asymptomatic individuals did not qualify for testing), study staff sent PCR sampling kits to subjects by post; completed test kits were then mailed to a central laboratory (Sanquin, Amsterdam, The Netherlands) for analysis. 941 self-tests were sent out, of which 731 were returned and analysed.

Additionally, all subjects were asked to take at-home capillary blood samples four times over the course of the study using finger pricks [20]: at baseline, and at the end of the learning phase, period 1, and period 2. Learning phase and period 2 samples underwent serology testing using in-house developed and validated total antibody assays [21,22] to determine whether a SARS-CoV-2 infection had occurred in the preceding interval. Seroconversion was confirmed by testing baseline or period 1 samples in case of a positive test after the learning phase or period 2, respectively. Initially, antibodies against the SARS-CoV-2 spike protein were assessed (anti-S serology test), but this approach cannot discriminate between infection- and vaccination-induced antibodies [23]. With the removal of COVID-19 vaccination as an exclusion criterion, a test detecting anti-nucleocapsid protein (anti-N serology) had to be used, because anti-N antibodies are elicited by infection only. Both tests have good concordance [22].

## Algorithm development

The first version of the experimental algorithm used a recurrent neural network (RNN) with two hidden layers based on Long Short Term Memory (LSTM) units. The algorithm leveraged time series data to detect deviations in physiological parameters compared to a healthy baseline. This version relied on data from 66 subjects who tested positive, either through PCR or serology testing, for SARS-CoV-2 in the COVI-GAPP study [10], which used the Ava bracelet on a sample of subjects from Liechtenstein. The algorithm was then enhanced using data from period 1 of the COVID-RED trial to develop version 2, which was applied in real-time during period 2. This iteration of the algorithm only included data from positive PCR tests during period 1 and not from serology tests, as results of those serology tests were not available in time for the algorithm to be released at the start of period 2. Refinement of the algorithm itself was done by investigating additional features and incorporating transfer learning. The algorithm would calculate the probability of infection for every participant on a given day and an alert would be given when this probability exceeded a specific threshold. Sensitivity was prioritised over specificity in deciding this threshold to be able to, amongst others, better detect asymptomatic infections.

## Outcomes

The primary endpoints of the study were app-provided, real-time, daily indications of potential SARS-CoV-2 infections, and diagnostic SARS-CoV-2 infection status as determined by self-reported positive SARS-CoV-2 test and/or serology results during follow-up.

## Statistical analysis

Analyses were performed on different analysis sets defined in the statistical analysis plan and evaluated in a Blinded Data Review Meeting. The intention-to-treat (ITT) set included all subjects randomised to one of the two study sequences. The efficacy analysis (EA) set included all ITT individuals who did not report a SARS-CoV-2 infection before the start of period 2, submitted all necessary serology samples, and were at least 80% compliant with wearable syncing *and* daily symptom diary completion during period 2. For the partial compliance (PC) set, the same criteria were applicable, except that people were also included if they were at least 80% compliant in *either* bracelet wearing *or* submitting the daily symptom diary. A Safety Analysis set was defined as anyone in the ITT that wore the bracelet at least once during the study to characterise the frequency and characteristics of reported adverse events. The primary analyses were performed on the EA and PC analysis sets.

Four primary analyses were conducted, which evaluated different aspects of the primary objective: time-to-infection; time-to-indication; ever-infected; and per-day. All analyses were performed using R statistical software version 4.1.2 [24].

**Time-to-infection analysis.** The time-to-infection analysis aimed to test the hypothesis that infection occurred at similar rates across groups and that being in either study condition did not change individuals' risk of getting infected or the likelihood of seeking a test in case of infection. The date of a first laboratory-confirmed SARS-CoV-2 infection (determined through self-reports in the Ava COVID-RED app, biweekly surveys, provided PCR self-sampling kits or periodic serology tests) was used as the clinical endpoint. Time until this date was compared between study conditions using a stratified log-rank test which assessed whether hazard functions were equal between groups.

**Time-to-indication analysis.** In the time-to-indication analysis, within-person time-to-indication was compared between study conditions by applying both algorithms to the same individual in the week prior to infection. Only subjects in the experimental condition in period 2 with a first-time SARS-CoV-2 infection, as confirmed through a SARS-CoV-2 test, were included in this analysis. The clinical endpoint of interest was the first red alert indicator in the week prior to the date at which the SARS-CoV-2 infection was confirmed through testing. The indications provided to the infected subjects in the week prior to their infection were compared to the predictions they would have received if they had been in the control condition such that it could be assessed how early both algorithms were able to detect an incoming infection. A Wilcoxon signed rank test was used to assess the significance of differences.

**Ever-infected analysis.** The ever-infected analysis assessed condition-based differences in the algorithms' performance to detect if a SARS-CoV-2 infection occurred during period 2. The number of subjects with at least one reported infection was cross-tabulated with the number of subjects with at least one "red alert" indication. The infection status in period 2 was considered positive if: (1) the subject reported a positive PCR or antigen test during period 2; or, (2) the serology test at the end of period 2 was positive while it was negative at the end of period 1. The indication status of a subject was considered positive if the subject received at least one red alert during period 2. For both study conditions, the performance of the algorithm was assessed based on the agreement between infection and indication status. This was done through calculation of the positive predictive value (PPV), negative predictive value (NPV), sensitivity, and specificity.

**Per-day analysis.** The per-day analysis compared the performance of both algorithms for detecting symptomatic SARS-CoV-2 infections per day of period 2. Only days for which a SARS-CoV-2 indication status was provided to the subject were included. Two different definitions were applied for infection status. In definition 1 (diagnostic test results only), a subject was considered SARS-CoV-2 positive three days prior to self-reported symptom onset until and including the first day of symptom onset; they were considered SARS-CoV-2 negative all other days. Symptom onset was defined as the day on which any COVID-19 symptoms were logged in the Ava COVID-RED app in combination with a positive SARS-CoV-2 test result a maximum of 14 days later. In definition 2 (addition of serology tests), subjects with a positive serology test at the end of period 2 and a negative test at the end of period 1 were also included as infections in the analysis. For those cases, the first day during period 2 on which COVID-19 associated symptoms were logged in the Ava COVID-RED app was considered the day of symptom onset. As with definition 1, the positive SARS-CoV-2 period was 3 days before symptom onset until the day of symptom onset. The indication status was the daily alert status generated by the app. The analysis was performed for both definitions separately.

All days for which both infection and indication status were known were classified into one of four outcomes: true positive (TP, both indication and infection status positive), false positive (FP, positive indication status with a negative infection status), true negative (TN, both indication and infection status negative), and false negative (FN, negative indication status with a positive infection status). This resulted in a total number of days that subjects were classified into one of the four categories, which could be used to calculate sensitivity, specificity and accuracy.

## Results

Between 22 February and 3 June 2021, 57,161 subjects were screened, and 17,825 fulfilled inclusion criteria to be randomised (Fig 3). Most randomised subjects (n = 10,822) were considered "normal risk", while 7,003 were considered "high-risk". 511 adverse device effects were reported of which four severe, but none serious (S1 Table in S1 File). During a Blinded Data Review Meeting, it was decided to also use a 60% compliance threshold to generate EA and PC analysis sets in addition to the a priori specified 80% compliance threshold, given the observed compliance rates (16% and 21% compliant in EA set for 80% and 60% threshold respectively; 22% and 26% compliant in PC set for 80% and 60% respectively). In this paper, we report results for the 60% compliance threshold given its higher external validity; results for the 80% compliance threshold can be found in the S3–S10 Tables and S4–S8 Figs in the S1 File.

Applying the 60% compliance threshold, the numbers of subjects in the EA and PC analysis sets were 3,811 and 4,619, respectively. The following analyses report the primary analysis results using these analysis sets and version 2 of the algorithm, which was applied in real-time during period 2.

Table 1 shows demographics and baseline characteristics for the EA and PC sets. The mean age of subjects was approximately 51 across analysis sets and study conditions, and the majority of subjects were females. Further baseline characteristics are presented in S2 Table in S1 File.

The time-to-infection analysis only included subjects who reported a first-time SARS-CoV-2 infection during period 2. In the EA set, these were 110 (5.8%) subjects for the control condition, and 129 (6.7%) for the experimental condition. In the PC set, these numbers were 143 (6.3%) and 162 (6.9%). Test statistics and p-values of the stratified log-rank test comparing time-to-infection between study conditions are shown in Table 2. The null hypothesis was not rejected for either of the analysis sets, suggesting that the experimental condition did not affect the time until confirmed infection. S1 and S2 Figs in S1 File present the corresponding Kaplan-Meier curves.

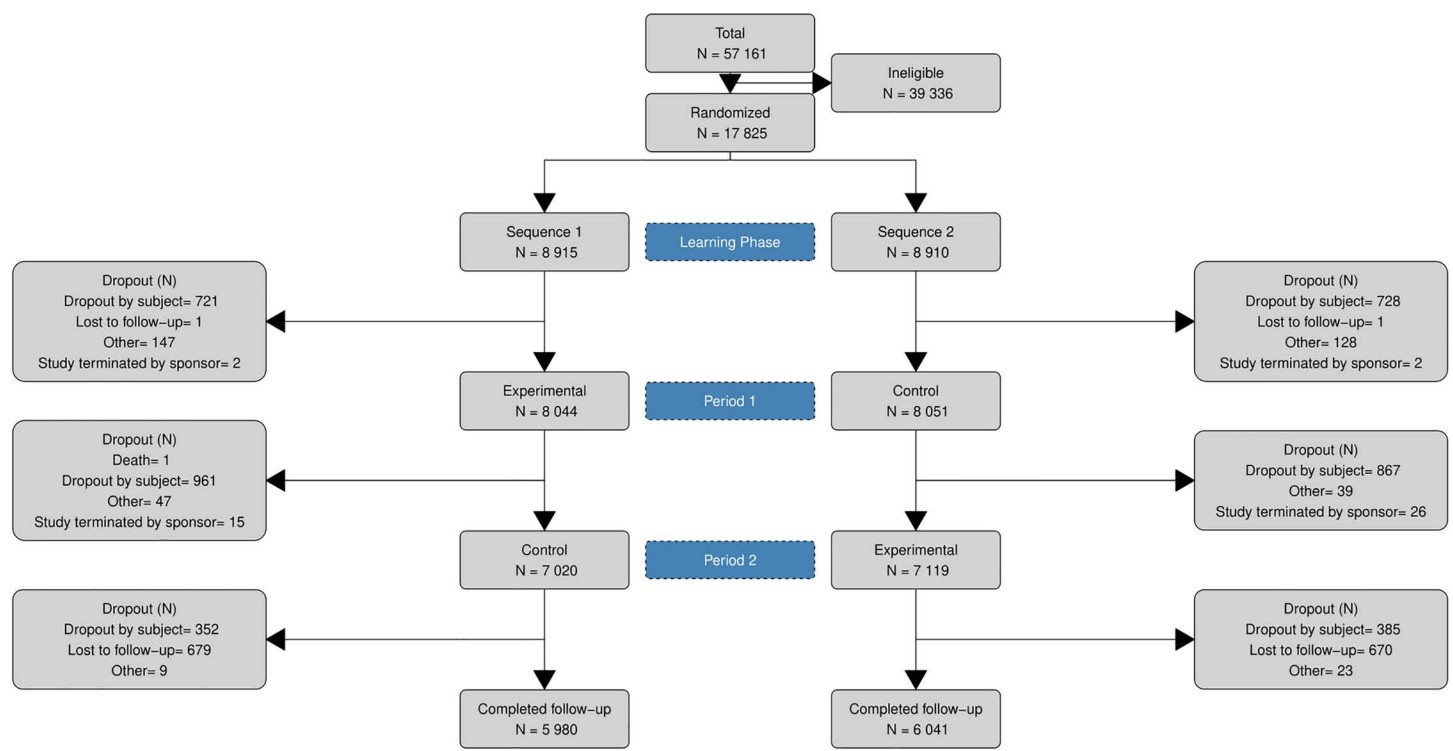

**Fig 3. CONSORT diagram of the COVID-RED trial.**

**Table 1. Baseline characteristics of the 60% compliance EA and PC analysis sets.**

| | | EA set | | PC set | |
| --- | --- | --- | --- | --- | --- |
| | | Control N=1,891 | Experimental N=1,920 | Control N=2,286 | Experimental N=2,333 |
| Age (years) | Mean (SD) | 51.3 (12.9) | 51.7 (12.6) | 50.9 (13.1) | 51.2 (12.9) |
| | Median (IQR) | 52.0 (43.0, 61.0) | 53.0 (44.0, 61.0) | 52.0 (42.2, 61.0) | 52.0 (43.0, 60.0) |
| Sex | Male | 517 (27.3%) | 519 (27.0%) | 627 (27.4%) | 639 (27.4%) |
| | Female | 1,374 (72.7%) | 1,401 (73.0%) | 1,659 (72.6%) | 1,649 (72.6%) |
| Risk group | High risk | 785 (41.5%) | 784 (40.8%) | 946 (41.4%) | 948 (40.6%) |
| | Normal risk | 1106 (58.5%) | 1136 (59.2%) | 1340 (58.6%) | 1385 (59.4%) |
| Body mass index (BMI) | Mean (SD) | 26.6 (4.9) | 26.6 (4.9) | 26.7 (4.9) | 26.7 (5.0) |
| Medical history | Any risk factor | 603 (31.9%) | 609 (31.7%) | 728 (31.8%) | 713 (30.6%) |
| Medication use | Any medication use | 805 (42.6%) | 808 (42.1%) | 961 (42.0%) | 958 (41.1%) |

**Table 2. Results of the time-to-infection analyses.**

| | EA set | | PC set | |
| --- | --- | --- | --- | --- |
| | Control N=1,891 | Experimental N=1,920 | Control N=2,286 | Experimental N=2,333 |
| Number of subjects with first-time infection | 110 (5.8%) | 129 (6.7%) | 143 (6.3%) | 162 (6.9%) |
| Hazard ratio experimental vs. control | | 0.86 | | 0.84 |
| p-value of stratified log-rank test | | 0.24 | | 0.34 |

The time-to-indication analysis included subjects from the experimental condition with a first-time SARS-CoV-2 infection during period 2. This resulted in a sample size of 27 for the EA set and 30 for the PC set. The within-person comparison in time-to-indication between study conditions led to the same conclusion in both analysis sets. Namely, subjects infected during the experimental condition received a positive indication significantly earlier compared to those infected during the control condition, with a median of 0 versus 7 days prior to the positive SARS-CoV-2 test. Table 3 shows the results of this analysis, with corresponding Kaplan-Meier plots in S3 and S4 Figs in S1 File.

The ever-infected analysis assessed the performance of the algorithms for detecting infections during period 2. In the EA set, 67.7% of subjects received a red alert at least once, with 6.3% testing positive. The percentages are 65.2% and 6.6%, respectively, in the PC set. Table 4 shows a cross-tabulation of infection versus indication for both analysis sets. In the EA set, the experimental algorithm achieved 99.2% sensitivity and 0.8% specificity, while the control algorithm achieved 46.4% sensitivity and 65.0% specificity. In the PC set, the experimental algorithm achieved 93.8% sensitivity and 4.2% specificity, while the control algorithm achieved 43.4% sensitivity and 66.4% specificity. Thus, the experimental

**Table 3. Time-to-indication and corresponding p-values.**

| | | EA set N=27 | | PC set N=30 | |
| --- | --- | --- | --- | --- | --- |
| | | Control | Experimental | Control | Experimental |
| Time-to-indication (days prior to positive SARS-CoV-2 test) | Minimum | 3 | 7 | 3 | 7 |
| | Median | 0 | 7 | 0 | 7 |
| | Maximum | 0 | 0 | 0 | 0 |
| p-value of Wilcoxon signed-rank test | | | < 0.001 | | < 0.001 |

algorithm was able to identify most infections, but generated many false positive indications. The control algorithm, on the other hand, detected fewer than half of the infections, but also generated fewer false positives. The NPV and PPV did not differ significantly between study conditions, although both were slightly higher in the control condition (S3 Table in S1 File).

The aim of the per-day analysis was to determine the experimental and control algorithms' likelihood of detecting a SARS-CoV-2 infection on a given day. Subjects who discontinued within one month after the start of period 2, or had a positive SARS-CoV-2 infection test within five days after the start of period 2, were excluded from the per-day analysis. Table 5 shows the number of included subjects and days that were used for analyses, as well as the measures of interest for both definitions.

The experimental algorithm achieved higher sensitivity than the control condition in both the EA and PC sets and for both definitions (45–52% versus 28–33%), but much lower specificity (38–50% versus 93–97%). The accuracies of the experimental algorithm were also much lower compared to those of the control algorithm. Experimental algorithm sensitivity was lower when self-reported test results were used without serology results, while the opposite held for the control algorithm (Table 5). Specificity was higher when serology was included for both algorithms.

## Discussion

Results of this study show that alerts based on both physiological data and self-reported symptoms were given significantly earlier than those based solely on self-reported symptoms, but this increased alert rate came at the cost of increased false positive rates. Moreover, the experimental algorithm achieved high sensitivity when detecting SARS-CoV-2 infections during a specified period, but specificity was low. Similarly, for the detection of infections on a given day, the experimental algorithm achieved higher sensitivity than the control algorithm, but specificity was much lower for the algorithm ingesting wearable device data. This low specificity also influences the interpretation of the results of the early detection analysis. The experimental algorithm's tendency to generate many false positive alerts increased the likelihood of an alert on any given day, which in turn contributed to individuals in the experimental condition being alerted earlier than those in the control condition. Despite the complex interpretation of the results, the unprecedented scale of this study provided invaluable lessons on the development and evaluation of novel machine learning algorithms for infectious disease detection and a large multidisciplinary dataset that will facilitate future research in the domain.

This work builds on previous literature on the use of wearable devices for detecting SARS-CoV-2, which often lacked applicability in the real world due to retrospective study designs [9]. Some prognostic studies have been conducted, but these enrolled less generalisable subject populations [10], and did not include a control group [10,15–17]. The

**Table 4. Cross-tabulation of infection versus indication status for both analysis sets and study conditions.**

| | EA set | | PC set | |
| --- | --- | --- | --- | --- |
| | Infected | Not infected | Infected | Not infected |
| **Overall** | | | | |
| Indication positive | 179 | 2400 | 214 | 2798 |
| Indication negative | 60 | 1172 | 91 | 1516 |
| **Experimental condition** | | | | |
| Indication positive | 128 | 1777 | 152 | 2079 |
| Indication negative | 1 | 14 | 10 | 92 |
| **Control condition** | | | | |
| Indication positive | 51 | 623 | 62 | 719 |
| Indication negative | 59 | 1158 | 81 | 1424 |

**Table 5. Results of the per-day analysis for both definitions.**

**Results using definition 1 (diagnostic test results only)**

|  | EA | | PC | |
|---|---|---|---|---|
|  | Experimental | Control | Experimental | Control |
| **Number of subjects** | 6 | 7 | 6 | 7 |
| **Number of days** | 272 | 298 | 272 | 298 |
| **TP** | 9 | 9 | 9 | 9 |
| **TN** | 96 | 254 | 96 | 254 |
| **FP** | 156 | 17 | 156 | 17 |
| **FN** | 11 | 18 | 11 | 18 |
| **Sensitivity** | 45.0% | 33.3% | 45.0% | 33.3% |
| **Specificity** | 38.1% | 93.7% | 38.1% | 93.7% |
| **Accuracy** | 38.6% | 88.3% | 38.6% | 88.3% |

**Results using definition 2 (including serology)**

|  | EA | | PC | |
|---|---|---|---|---|
|  | Experimental | Control | Experimental | Control |
| **Number of subjects** | 17 | 19 | 20 | 21 |
| **Number of days** | 615 | 715 | 688 | 779 |
| **TP** | 33 | 21 | 37 | 23 |
| **TN** | 276 | 623 | 302 | 682 |
| **FP** | 275 | 18 | 312 | 18 |
| **FN** | 31 | 53 | 37 | 56 |
| **Sensitivity** | 51.6% | 28.4% | 50.0% | 29.1% |
| **Specificity** | 50.1% | 97.2% | 49.2% | 97.4% |
| **Accuracy** | 50.2% | 90.1% | 49.3% | 90.5% |

COVID-RED study was one of the first and largest randomised prospective studies to apply an algorithm based on physiological parameters for early detection of SARS-CoV-2 and to alert subjects in real-time, often before symptom onset. It was also unique in its use of serology testing in addition to self-reported test results. The COVID-RED study was conducted during an ongoing pandemic, thereby making it representative of a real-world scenario where wearable devices are used for tracking changes in physiological parameters.

While the experimental algorithm achieved high sensitivity in the ever-infected and per-day analyses, and shorter time-to-indication than the symptom-only algorithm, the algorithm generated numerous false positive alerts, which resulted in very low specificity. This might be partially explained by the algorithm's inability to differentiate between SARS-CoV-2 and other (respiratory) infections. Moreover, the algorithm was, amongst others, developed with the aim to detect asymptomatic infections, which informed decisions in algorithm development to prioritise sensitivity over specificity. Partly due to the low specificity, an economic evaluation of the trial indicated that the use of the wearable device and the experimental algorithm in the general population would likely not be cost-effective [25]. Results of the study remain relevant for limiting virus spread and could potentially be of use for early treatment of disease. Further research can look into finetuning the algorithm and improve its specificity, while also evaluating the potential of using wearable device data for detecting influenza and viral diseases in general. The possibility of doing this has already been discussed in the literature [26,27]. As detailed in the methods, we developed a third version of the algorithm based on data from across all three study periods; its retrospective and iterative performance is beyond the scope of this paper but will be detailed in a forthcoming publication.

Even though the current paper focuses on the performance of the algorithm that was applied in real-time during the COVID-RED trial (version 2), we envision multiple ways in which the algorithm could be improved to achieve better specificity. A first suggestion would be to use additional methodologies, such as the Youden index [28], to determine a better cut-off point for the algorithm to generate red alerts. While this adjusted cut-off would lead to decreased sensitivity, the specificity and overall accuracy could be improved. Other machine learning methodologies to better balance sensitivity and specificity could also be considered. The cut-off point could also be improved using metrics like the AUC, which was not presented due to the inability to access proprietary model outputs. It is therefore not possible to compare the AUC to those achieved in previous studies investigating the use of a wearable device for detecting SARS-CoV-2 infections. However, given the algorithm's low specificity with the current decision threshold and the highlighted differences between the interventions, reporting and comparing the AUC would not influence the conclusion of this study. Additionally, we could collect more detailed infection data in the training dataset by testing for multiple viruses, not just for SARS-CoV-2. Finally, implementing continuous learning in the experimental algorithm might lead to improved performance. The algorithm was developed when much was still unknown about SARS-CoV-2 and conditions changed continuously over the course of the trial. The algorithm was frozen from the start of a period, with its training datasets limited to data previously collected. Thus, the algorithm could not be adapted to changing epidemiological settings without jeopardising the ability to compare its performance over time. Setting up the algorithm from the start for continuous learning could have enabled dynamic updating in a way that best reflected changing settings. Such a set-up would also have allowed for implementing dynamic cut-off points for generating red alerts, which could adapt to the epidemiological context at any point in time.

Many of the current study's limitations relate to the ever-changing environment in which it was performed. At the time of study, much was still unknown about SARS-CoV-2 and the epidemiological setting was constantly evolving. For instance, this study only investigated first-time infections, and subjects were not followed up after reporting a first-time infection due to the assumption that people could only get infected with SARS-CoV-2 once. However, it is now widely known that individuals can get infected multiple times, with the likelihood of re-infection increasing with more recent variants [29,30]. With that knowledge, an algorithm with a high per-day sensitivity and specificity that can detect any infection, regardless of potential previous infections, would be more applicable in a real-world scenario. The ever-changing environment also meant that, while initially it was assumed that only anti-S serology tests would be needed, anti-N serology tests had to be added during the study due to the widespread uptake of vaccination. This could have introduced measurement biases, although a systematic review identified no significant difference in sensitivity and specificity between both tests [31].

Compliance to study conditions could also be considered a limitation of this study. Even after lowering the compliance threshold from 80% to 60%, the number of participants included in the main analyses was much lower than the number of participants initially randomised. However, given the unique circumstances and the decentralised nature of this study, it is extremely difficult to determine what a realistic compliance rate would have been. Over the course of the study, additional compliance interventions were tested [32].

A final limitation is that the exact timing of infection could not be determined from only positive serology tests. Because of this, it was not always clear whether a red alert and reported symptoms pertained to the same infection, which introduced uncertainty into several primary analyses. For example, in the time-to-infection analysis, the inclusion of serology testing, through which most infections were detected, meant that finding differences between the study conditions in time-to-infection was more challenging. Future research can investigate alternative approaches and evaluate the time-to-infection in more detail.

## Conclusions

The COVID-RED study was the largest wearable device study during the course of the pandemic, enrolling over 17,000 subjects. Despite its establishment in the early days of the COVID-19 pandemic and the ever-changing epidemiological and societal context, the study findings may serve as a prelude to the potential future role of wearable devices in

infectious disease surveillance. Currently the experimental algorithm achieved high sensitivity at the cost of low specificity. Further research could look into finetuning the algorithm to improve specificity, or into repurposing the algorithm to serve for detecting respiratory disease in general. The large amount of valuable data collected in the COVID-RED study were made publicly available [33], with the hope that its publication will contribute to further research on SARS-CoV-2 and provide a unique wearable-based repository for future scientific inquiry.

## Supporting information

**S1 File. Supplementary tables and figures.**
(DOCX)

**S2 File. CONSORT checklist.**
(DOCX)

**S3 File. Protocol document.**
(PDF)

## Acknowledgments

**Consortia. On behalf of the COVID-RED consortium:** The contributors associated with COVID-19 Remote Early Detection (COVID-RED) consortium are as follows: Maureen Cronin, Vladimir Kovacevic, Andjela Markovic, Maja Rudinac, from Ava AG, Switzerland; Kirsten Grossmann, Lorenz Risch, Martin Risch, Ornella Weideli, from Dr. Risch, Liechtenstein; Billy Franks, Brianna Goodale, Ellen Dutman, Eric Houtman, Glenn Van Wigcheren, Hans Van Dijk, Ishak Elmouhajir, Jon Bouwman, Lotte Smets, Marcel van Willigen, Niki de Vink, Timo Brakenhoff, Titia Leurink, Wendy van Scherpenzeel, Wout Aarts, Pieter van der Meer, Myrna Verhulst, Paul Klaver, Tessa Heikamp, Kai Hage, José Broersen, Jungyeon Choi, Maartje Hoffmann, Marjolein Jansen, Jeffrey Burggraaff, Laura Zwiers, from Julius Clinical, The Netherlands; Alison Kuchta, Christian Simon, Santiago Montes, from Roche, The Netherlands; Theo Rispens, Maurice Steenhuis, Floris Loeff, Sofie Keijzer, Jim Keijser, Olvi Christianawati, Aren Boogaard, Nadine Commandeur, from Sanquin, The Netherlands; Ariel Dowling and Steve Emby, from Takeda, USA; Charisma Hehakaya, Daniel Oberski, George Downward, Duco Veen, Marianna Mitratza, Hans Reitsma, Janneke van de Wijgert, Nathalie Vigot, Patricia Bruijning, Pieter Stolk, Diederick Grobbee*, Gulseren Yalvac, from University Medical Center Utrecht, The Netherlands; Johann Fevrier, Amos Folarin, Pablo Fernandez Medina, Richard Dobson, Spiros Denaxas, from University College London, UK; Eskild Fredslund, Serkan Korkmaz, and Jesper Strømstad, from VIVE, Denmark. *Diederick Grobbee is lead author of this group (d.e.grobbee@umcutrecht.nl).

## Author contributions

**Conceptualization:** Billy Franks, Lorenz Risch, David Conen, Martin Risch, Maureen Cronin, Diederick E. Grobbee.

**Data curation:** Timo B. Brakenhoff, Brianna Mae Goodale, Vladimir Kovacevic, Andjela Markovic, Marcel van Willigen, Serkan Korkmaz, Jakob Kjellberg, Lorenz Risch, Martin Risch, Kirsten Grossman, Theo Rispens, Maureen Cronin.

**Formal analysis:** Timo B. Brakenhoff, Duco Veen, Vladimir Kovacevic, Andjela Markovic, Marianna Mitratza, Kirsten Grossman, Amos A. Folarin, Xi Bai.

**Funding acquisition:** Maureen Cronin, Diederick E. Grobbee.

**Investigation:** Timo B. Brakenhoff, Diederick E. Grobbee.

**Methodology:** Timo B. Brakenhoff, Brianna M. Goodale, Duco Veen, Andjela Markovic, Janneke van de Wijgert, Maureen Cronin, Diederick E. Grobbee.

**Project administration:** Timo B. Brakenhoff, Brianna M. Goodale, George S. Downward, Lorenz Risch, David Conen, Martin Risch, Kirsten Grossman, Ornella C. Weideli, Jon Bouwman, Richard Dobson.

**Resources:** Theo Rispens.

**Supervision:** Diederick E. Grobbee.

**Writing – original draft:** Laura C. Zwiers, Timo B. Brakenhoff, Brianna M. Goodale, Duco Veen, George S. Downward.

**Writing – review & editing:** Laura C. Zwiers, Timo B. Brakenhoff, Brianna M. Goodale, Duco Veen, George S. Downward, Vladimir Kovacevic, Andjela Markovic, Marianna Mitratza, Marcel van Willigen, Billy Franks, Janneke van de Wijgert, Santiago Montes, Serkan Korkmaz, Jakob Kjellberg, Lorenz Risch, David Conen, Martin Risch, Kirsten Grossman, Ornella C. Weideli, Theo Rispens, Jon Bouwman, Amos A. Folarin, Xi Bai, Richard Dobson, Maureen Cronin, Diederick E. Grobbee.

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
