## [Decision Letter · Decision Letter 0]

5 Dec 2024

PONE-D-24-38787A wearable-based AI algorithm for the remote early detection of SARS-CoV-2 infections: results from the COVID-RED study, a prospective randomised single-blinded crossover trialPLOS ONE

Dear Dr. Zwiers,

Thank you for submitting your manuscript to PLOS ONE. After careful consideration, we feel that it has merit but does not fully meet PLOS ONE’s publication criteria as it currently stands. Therefore, we invite you to submit a revised version of the manuscript that addresses the points raised during the review process.

We look forward to receiving your revised manuscript.

Kind regards,

Cecilia Acuti Martellucci, M.D.

Academic Editor

PLOS ONE

Journal Requirements:

2. We note that the current version ethics document has masked details. Before we proceed with your submission, please upload a clean version of the ethics document. 

3. During our internal review of your submission, we noted that your clinical trial registration number (NL9320) cannot be found on the the Netherlands Trial Register. Please provide the correct registration number and a link to the online registration page if possible. Thank you for your attention this request. 

4. Please note that funding information should not appear in the Acknowledgments section or other areas of your manuscript. We will only publish funding information present in the Funding Statement section of the online submission form. Please remove any funding-related text from the manuscript. 

“I have read the journal's policy and the authors of this manuscript have the following competing interests: 

BMG, VK, AM, and MC are current or previous employees of Ava AG.

LCZ, TBB, BMG, DV, MW, BF, JB, DEG are current or previous employees of Julius Clinical.

BF is currently an employee of Haleon.

KG and OCW are current or previous employees of Dr Risch.

MR and LR are current employees and key shareholders at Dr Risch.”

We note that one or more of the authors are employed by a commercial company: Ava AG, Julius Clinical, Haleon, Dr Risch

2) Please also provide an updated Competing Interests Statement declaring this commercial affiliation along with any other relevant declarations relating to employment, consultancy, patents, products in development, or marketed products, etc.  

Within your Competing Interests Statement, please confirm that this commercial affiliation does not alter your adherence to all PLOS ONE policies on sharing data and materials by including the following statement: ""This does not alter our adherence to PLOS ONE policies on sharing data and materials.” (as detailed online in our guide for authors http://journals.plos.org/plosone/s/competing-interests). 

If this adherence statement is not accurate and there are restrictions on sharing of data and/or materials, please state these. Please note that we cannot proceed with consideration of your article until this information has been declared.

6. One of the noted authors is a group or consortium: COVID-RED consortium

In addition to naming the author group, please list the individual authors and affiliations within this group in the acknowledgments section of your manuscript. Please also indicate clearly a lead author for this group along with a contact email address.

7. Please include a caption for figure 1. 

8. Please upload a new copy of Figure 2 as the detail is not clear. Please follow the link for more information:

https://blogs.plos.org/plos/2019/06/looking-good-tips-for-creating-your-plos-figures-graphics/

https://blogs.plos.org/plos/2019/06/looking-good-tips-for-creating-your-plos-figures-graphics/

10. We note that the original protocol file you uploaded contains a confidentiality notice indicating that the protocol may not be shared publicly or be published. Please note, however, that the PLOS Editorial Policy requires that the original protocol be published alongside your manuscript in the event of acceptance. Please note that should your paper be accepted, all content including the protocol will be published under the Creative Commons Attribution (CC BY) 4.0 license, which means that it will be freely available online, and any third party is permitted to access, download, copy, distribute, and use these materials in any way, even commercially, with proper attribution.

Therefore, we ask that you please seek permission from the study sponsor or body imposing the restriction on sharing this document to publish this protocol under CC BY 4.0 if your work is accepted. We kindly ask that you upload a formal statement signed by an institutional representative clarifying whether you will be able to comply with this policy. Additionally, please upload a clean copy of the protocol with the confidentiality notice (and any copyrighted institutional logos or signatures) removed.

**Additional Editor Comments:**

The submitted manuscript reports on a rigorous and extensive work. My major concern is the inconsistency between the results and the conclusions. The findings will be much more useful for future research if they are commented faithfully, without claims that are not supported by the data. Please ensure that this is revised throughout the whole manuscript.

Reviewers' comments:

Reviewer's Responses to Questions

**Comments to the Author**

1. Is the manuscript technically sound, and do the data support the conclusions?

Reviewer #1: Yes

Reviewer #2: Yes

Reviewer #3: No

Reviewer #4: No

2. Has the statistical analysis been performed appropriately and rigorously? 

Reviewer #1: Yes

Reviewer #2: Yes

Reviewer #3: Yes

Reviewer #4: Yes

3. Have the authors made all data underlying the findings in their manuscript fully available?

Reviewer #1: Yes

Reviewer #2: Yes

Reviewer #3: Yes

Reviewer #4: Yes

4. Is the manuscript presented in an intelligible fashion and written in standard English?

Reviewer #1: Yes

Reviewer #2: Yes

Reviewer #3: Yes

Reviewer #4: Yes

5. Review Comments to the Author

Reviewer #1: The trial design is quite interesting, as it involves a crossover with two groups over two periods. The statistical analysis is well-defined and rigorously applied, and the results are well-reported. However, the specificity of the experimental protocol was notably low and disappointing. This has important implications, as a protocol that very frequently returns positive results tends to overestimate infections. While it is preferable to report an infection rather than miss it, the extremely low specificity raises concerns. I am resistant to say that a device would necessarily be effective or (cost effective) in these circumstances. The manuscript does point to the low specificity and says that future finetuning mechanisms are needed. I would like to see more discussion on how to do finetuning and to evaluate the effectiveness.

An initial suggestion would be to use the Youden index to establish an appropriate cutoff, ensuring the protocol is more effective in practice, although finetuning most likely involves more effort than simply using this index.

Other comments and suggestions are provided below.

Abstract

- The phrase "Performance evaluated using measures of diagnostic accuracy" is vague. It would be helpful to provide more specific details on the evaluation metrics used.

- The reported specificity of "0.8–4.4%" seems likely to be a typo. I believe it should be "0.8%–4.2%." Please verify this.

- The term "algorithm ingesting data" is somewhat technical (my impression) for an abstract. To reach a broader audience, I suggest rephrasing it to "data was used as input to the algorithm" or something along these lines.

Introduction

- "Relatively young sample of individuals" should be revised to "sample of relatively young individuals."

- Line 194: Correct the phrase to "daily diary."

Methodology

- Was a sample size analysis performed? While the number of people enrolled seems sufficiently large, there is a question about whether the study period was sufficient to observe a number of outcomes. I recommend including information on sample size analysis conducted.

- The description of "per-day analysis" is unclear. I understand it as calculating the number of infected days across all participants, which serves as a measure of person-time. Please verify if this interpretation is correct and consider clarifying the description in the text and in Table 5.

- Another suggestion is to report the Youden index (sensitivity + specificity - 1) and indicate under which circumstances this index was positive (or exceeded a defined cutoff).

Results

- Figure 2 requires higher image resolution or better text contrast. When I attempted to download it to try to see in better resolution, only the Figure 3 was downloaded. I double checked it but I am not sure why this happens.

- In S1 Appendix, I suggest adding to the captions that "survival probability" refers to the probability that the event has not occurred.

Discussion

As indicated above, I would like to see a clearer outline of how to fine-tune the protocol, as well as metrics for a cost-effectiveness analysis.

Reviewer #2: Main strength's of the paper:

1) Innovative concept and relevance

The COVID-RED study embodies an innovative approach by leveraging wearable technology and artificial intelligence (AI) to address a critical public health issue: the early detection of SARS-CoV-2 infections. This research is particularly relevant in light of the COVID-19 pandemic, where the ability to identify infections during the pre-symptomatic or asymptomatic phase could significantly curb transmission rates. The study bridges advanced AI algorithms with wearable medical devices, showcasing a futuristic and interdisciplinary method for monitoring public health. This integration of technology with healthcare represents a paradigm shift from traditional diagnostics to more proactive and predictive health monitoring.

2) Robust study design

The study design is meticulously structured to ensure reliability and scientific rigor:

- Randomized Controlled Trial (RCT): A prospective, single-blinded, two-period, two-sequence, randomized crossover trial was employed. This robust design minimizes biases and allows a direct comparison between the experimental (wearable-based AI algorithm) and control (symptom-based) conditions.

- Large Sample Size: The inclusion of 17,825 participants is a remarkable achievement, providing high statistical power and the ability to generalize findings across populations.

- Crossover Nature: The two-period crossover design ensures that each participant serves as their control, reducing variability and enhancing the internal validity of the results.

These features collectively underscore the study's scientific robustness and provide a solid foundation for its conclusions.

3) Use of cutting-edge wearable technology

The article highlights the use of the Ava bracelet, an FDA-cleared and CE-certified wearable device originally designed for fertility tracking, repurposed for infection detection. The bracelet's ability to measure multiple physiological parameters, including respiratory rate, heart rate, heart rate variability, wrist-skin temperature, and skin perfusion, is a testament to its advanced capabilities.The study emphasizes the bracelet’s potential for continuous health monitoring, enabling real-time alerts for possible infections.Repurposing existing technology for public health surveillance is a forward-thinking approach that maximizes the utility of available resources.The wearable is user-friendly, requiring participants to wear it during sleep and synchronize data via a mobile app, ensuring convenience and accessibility.

4) Significant findings supporting early detection

The article demonstrates that the experimental wearable-based AI algorithm outperforms symptom-only models in detecting SARS-CoV-2 infections. The algorithm achieved high sensitivity (93.8-99.2%) in detecting infections during the study period, ensuring that most infected individuals were identified. Alerts based on the wearable device were issued significantly earlier (median of 7 days before a positive test) compared to symptom-based alerts, which had no prior warning.These findings underscore the potential of wearable devices to serve as an early warning system, particularly valuable during infectious disease outbreaks.

5) Comprehensive Performance Evaluation

The article provides a detailed evaluation of the experimental algorithm's performance through various metrics:

- Sensitivity vs. Specificity: While acknowledging the trade-offs (high sensitivity but low specificity), the study emphasizes the algorithm’s strengths in detecting infections, particularly in pre-symptomatic phases.

- Multi-Faceted Analysis: Different analytical approaches—time-to-infection, time-to-indication, and per-day analyses—offer a comprehensive assessment of the algorithm’s utility.

6) Real-World Applicability

The study's context—conducted during the COVID-19 pandemic—adds real-world relevance to its findings. Unlike retrospective studies, this trial evaluated the algorithm’s effectiveness in real-time, simulating actual scenarios where timely detection is critical.By combining physiological data with laboratory-confirmed SARS-CoV-2 infections, the study achieves a holistic approach to infection detection.

Areas of improvement

1) Specific challenges

One of the study's key findings is the low specificity of the experimental algorithm (0.8–4.4%) compared to the symptom-only control algorithm (65–66.4%). While high sensitivity ensures most infections are detected, the high rate of false positives has significant implications. Frequent false alerts can overwhelm testing resources and create undue anxiety among users, reducing trust in the system.Excessive testing for false positives may strain public health systems, particularly during pandemics.

Suggestions:

- Incorporate additional parameters, such as behavioral data or exposure history, to refine the algorithm.

- Use advanced machine learning techniques, such as ensemble methods or Bayesian inference, to balance sensitivity and specificity better.

- Develop dynamic thresholds that adapt to local infection prevalence, reducing false positives in low-risk settings.

2) Algorithm adaptability

The study design required freezing the algorithm at the start of each study period, which limited its ability to adapt to changing epidemiological conditions or incorporate newly available data. The fixed algorithm could have reduced the model's effectiveness, especially as new variants of SARS-CoV-2 emerged or vaccination rates increased.

Suggestions:

- Implement continuous learning mechanisms to allow the algorithm to evolve in response to new data, such as emerging variants or shifting symptoms.

- Explore federated learning, where the algorithm learns from decentralized data sources without compromising user privacy, to increase adaptability.

3) Limited focus on specific infections

The algorithm was designed to detect SARS-CoV-2 infections but lacked the ability to differentiate between COVID-19 and other respiratory illnesses. Many false positives could result from physiological changes due to non-COVID-19 infections, such as the flu or common cold. The study did not explore the potential of wearable devices for broader infectious disease surveillance.

Suggestions:

- Expand the dataset to include physiological data from individuals with other respiratory infections.

- Train the algorithm to distinguish between SARS-CoV-2 and other illnesses, enhancing its diagnostic utility.

- Investigate the algorithm’s potential for detecting comorbid conditions or other diseases that impact similar physiological parameters.

4) Generalizability of findings

Although the study's sample size was large, its findings may have limited applicability to diverse populations:

- Demographic Limitations: Most participants were from the Netherlands, potentially limiting generalizability to populations with different demographics, healthcare systems, or COVID-19 prevalence rates.

- Age Range and Health Status: The study included primarily younger and healthier individuals, with fewer older or high-risk participants.

Suggestions:

- Conduct follow-up studies in diverse geographic locations and healthcare contexts to ensure broader applicability.

- Stratify results by demographic factors (e.g., age, gender, pre-existing conditions) to understand how the algorithm performs across subgroups.

Reviewer #3: This manuscript presents the COVID-RED study, which investigates the use of the Ava bracelet for the early detection of SARS-CoV-2 infections in real-time. The study is noteworthy for its scale, involving approximately 20,000 participants spanning diverse age groups and genders. The results demonstrate that integrating wearable data significantly enhances the sensitivity of COVID-19 detection, although it comes at the cost of reduced specificity compared to the control group without wearable data. The study is well-designed, and the dataset is particularly valuable due to its large sample size. The analyses are robust, and the manuscript is clearly written and well-structured.

However, my primary concern lies in the manuscript's alignment with its title, "A wearable-based AI algorithm for the remote early detection...". While the study focuses on the advantage of including wearable data, there is insufficient information about the AI algorithm itself. Details on the algorithm—both with and without wearable data—are largely absent. Since algorithmic performance can significantly impact predictive outcomes, it is essential to provide more information about the models used, their training processes, and any comparisons made to alternative algorithms. Without this critical component, it is challenging to assess whether the best-performing algorithms were employed for both experimental and control groups. I recommend including these details to enable a more comprehensive evaluation of the study.

Reviewer #4: Dear authors,

Thank you for your collective efforts to advance the use of wearable devices in infectious disease surveillance. Here are some points to consider regarding your paper:

There is no caption in the text for Figure 1. Other figures also seem to be incorrectly referenced in the body of text (1 for 2, 2 for 3, 3 for 1).

Line 220: A brief description of the ML algorithm used would be helpful.

Line 226: How were the participants who did not seek testing classified?

Line 229: How were the participants who did not return the kits classified? (completed follow-up/drop-out/…)

Line 232: repeated “the end of”

Line 309: How was the beginning of infection determined in asymptomatic cases?

Line 336: Table legends should be placed above the body of the table (not below).

Line 381: Please provide the value of AUC in addition to the stated metrics (line 316).

Line 384: The minimum and median time-to-indication for the experimental condition in both groups are seven days (Line 354). I see that it is stated in line 276, “The clinical endpoint of interest was the first red alert indicator in the week prior to the date at which the SARS-CoV-2 infection was confirmed through testing”. Here, the question is raised regarding the basis upon which this timeframe (one week or seven days) is determined. If a different duration was chosen, supposedly 3 days or 14 days, would we have seen the median to be 3 days and 14 days, respectively? Considering the high false positive rate of the model, more explanation could be helpful to differentiate it from a random model, stochastically raising alarms. Such a random model could have still achieved similar results of seven days, rendering the significance of this finding obsolete (Line 384).

Line 390: It is acknowledged that both sensitivity and specificity improved using the second definition. Although the experimental condition achieved a sensitivity superior to that of control, specificity was yet considerably lower. So, it would be incorrect to deduce that the experimental algorithm performed better “… in comparison to the control algorithm”. The accuracy is still lower than that of the control (50.2% Vs. 90.1%). A comparison cannot be based merely on sensitivity. It would be much more helpful if the area under the curve is presented for each algorithm as it was previously planned in the methods section (line 316) to help make a comparison.

Line 408: “This might be explained by the algorithm’s inability to differentiate between SARS-CoV-2 and other (respiratory) infections”. A much better explanation is provided on Line 422. Other infections, given that 67.7% of subjects received a red alert at least once, is a farfetched explanation.

Line 460: “… with the experimental algorithm achieving high sensitivity”. Partial representation of the findings-should be revised. Nonetheless, the study can “confirm a potential future role of wearable devices in infectious disease surveillance” even without an outstanding performance of the algorithm.

6. PLOS authors have the option to publish the peer review history of their article (what does this mean? ). If published, this will include your full peer review and any attached files.

**Do you want your identity to be public for this peer review?** For information about this choice, including consent withdrawal, please see our Privacy Policy .

Reviewer #1: No

Reviewer #2: **Yes: ** Axel Moyal

Reviewer #3: No

Reviewer #4: **Yes: ** Shahrokh Mousavi

---

## [Author Response · Author response to Decision Letter 1]

10 Feb 2025

Author response:

The revised version of the manuscript adheres to the indicated style requirements.

- We note that the current version ethics document has masked details. Before we proceed with your submission, please upload a clean version of the ethics document.

Author response:

A clean version of the ethics document has been uploaded.

- During our internal review of your submission, we noted that your clinical trial registration number (NL9320) cannot be found on the the Netherlands Trial Register. Please provide the correct registration number and a link to the online registration page if possible. Thank you for your attention this request.

Author response:

We realized that the website of the Netherlands Trial Register has been discontinued, which indeed makes it difficult to find the registered trial online. The trial number is still correct, and can be verified through the following two webpages: Remote Early Detection of SARS-CoV-2 infections | Research with human participants; ICTRP Search Portal.

- Please note that funding information should not appear in the Acknowledgments section or other areas of your manuscript. We will only publish funding information present in the Funding Statement section of the online submission form. Please remove any funding-related text from the manuscript.

Author response:

The funding-related text has been removed from the manuscript. Updated funding information is provided in the online submission form and in the cover letter.

- We note that one or more of the authors are employed by a commercial company: Ava AG, Julius Clinical, Haleon, Dr Risch

o Please provide an amended Funding Statement declaring this commercial affiliation, as well as a statement regarding the Role of Funders in your study. If the funding organization did not play a role in the study design, data collection and analysis, decision to publish, or preparation of the manuscript and only provided financial support in the form of authors' salaries and/or research materials, please review your statements relating to the author contributions, and ensure you have specifically and accurately indicated the role(s) that these authors had in your study. You can update author roles in the Author Contributions section of the online submission form.

o Please also include the following statement within your amended Funding Statement.

Author response:

The funder of this study (Innovative Medicines Initiative) did not provide support for authors in the form of salaries. Author salaries were paid by the various consortium partners that authors are affiliated with. We have therefore not fully included the above text in the amended Funding Statement. The statement does indicate that the funding body had no role in the design of the study; the collection, analysis, and interpretation of the data; and the writing of the manuscript. In addition, the funding statement indicates that the manuscript only reflects the personal views of the stated authors, and not those of the commercial companies they are affiliated with.

o If your commercial affiliation did play a role in your study, please state and explain this role within your updated Funding Statement.

o Please also provide an updated Competing Interests Statement declaring this commercial affiliation along with any other relevant declarations relating to employment, consultancy, patents, products in development, or marketed products, etc.

o Within your Competing Interests Statement, please confirm that this commercial affiliation does not alter your adherence to all PLOS ONE policies on sharing data and materials by including the following statement: ""This does not alter our adherence to PLOS ONE policies on sharing data and materials.” (as detailed online in our guide for authors http://journals.plos.org/plosone/s/competing-interests).

o If this adherence statement is not accurate and there are restrictions on sharing of data and/or materials, please state these. Please note that we cannot proceed with consideration of your article until this information has been declared.

o Please include both an updated Funding Statement and Competing Interests Statement in your cover letter. We will change the online submission form on your behalf.

Author response:

The Funding Statement and Competing Interest Statements have been updated following the editor’s instructions. Updated versions are in the cover letter.

- One of the noted authors is a group or consortium: COVID-RED consortium

o In addition to naming the author group, please list the individual authors and affiliations within this group in the acknowledgments section of your manuscript. Please also indicate clearly a lead author for this group along with a contact email address.

Author response:

Members of the COVID-RED consortium are now listed in the acknowledgements section of the manuscript. The lead author of this group is indicated in line 521.

- Please include a caption for figure 1.

Author response:

A caption is included in the revised manuscript. The figure number was adjusted, and this is now figure 3. Other figure numbers have been adjusted accordingly.

- Please upload a new copy of Figure 2 as the detail is not clear.

Author response:

A new copy is uploaded with the revised manuscript.

- Please include captions for your Supporting Information files at the end of your manuscript, and update any in-text citations to match accordingly.

Author response:

Captions and updated in-text citations were included in the revised manuscript.

- We note that the original protocol file you uploaded contains a confidentiality notice indicating that the protocol may not be shared publicly or be published. Please note, however, that the PLOS Editorial Policy requires that the original protocol be published alongside your manuscript in the event of acceptance. Please note that should your paper be accepted, all content including the protocol will be published under the Creative Commons Attribution (CC BY) 4.0 license, which means that it will be freely available online, and any third party is permitted to access, download, copy, distribute, and use these materials in any way, even commercially, with proper attribution.

o Therefore, we ask that you please seek permission from the study sponsor or body imposing the restriction on sharing this document to publish this protocol under CC BY 4.0 if your work is accepted. We kindly ask that you upload a formal statement signed by an institutional representative clarifying whether you will be able to comply with this policy. Additionally, please upload a clean copy of the protocol with the confidentiality notice (and any copyrighted institutional logos or signatures) removed.

Author response:

The study protocol is public domain in compliance with EU regulations and has been previously published, as indicated in line 145 of the revised manuscript. The currently uploaded version of the protocol file does not contain a confidentiality notice and can be published alongside the manuscript in the event of acceptance.

Additional Editor comments

The submitted manuscript reports on a rigorous and extensive work. My major concern is the inconsistency between the results and the conclusions. The findings will be much more useful for future research if they are commented faithfully, without claims that are not supported by the data. Please ensure that this is revised throughout the whole manuscript.

Author response:

We thank the editor for their compliments on the work performed. Textual adjustments have been made throughout the revised manuscript to avoid any claims that are not supported by the data.

Reviewer 1

The trial design is quite interesting, as it involves a crossover with two groups over two periods. The statistical analysis is well-defined and rigorously applied, and the results are well-reported. However, the specificity of the experimental protocol was notably low and disappointing. This has important implications, as a protocol that very frequently returns positive results tends to overestimate infections. While it is preferable to report an infection rather than miss it, the extremely low specificity raises concerns. I am resistant to say that a device would necessarily be effective or (cost effective) in these circumstances. The manuscript does point to the low specificity and says that future finetuning mechanisms are needed. I would like to see more discussion on how to do finetuning and to evaluate the effectiveness.

An initial suggestion would be to use the Youden index to establish an appropriate cutoff, ensuring the protocol is more effective in practice, although finetuning most likely involves more effort than simply using this index.

Author response:

We thank the reviewer for their kind words about the manuscript. We agree that the extremely low specificity of the algorithm is a limitation to the study. However, the current study’s scope was to report on the performance of the algorithm that was applied in real-time during the trial, which resulted in a high number of false positives. Further research, including also the forthcoming paper mentioned in Line 427 of the manuscript, will focus on methods for improving the algorithm. We have also shared as much data from this study as possible so that others can learn from our experiences.

We have added some further discussion (e.g., line 430) on how to finetune the algorithm in the future. Cost-effectiveness was discussed in another (preprint) publication, which has now been referenced in the revised discussion (line 420).

Abstract

- The phrase "Performance evaluated using measures of diagnostic accuracy" is vague. It would be helpful to provide more specific details on the evaluation metrics used.

- The reported specificity of "0.8–4.4%" seems likely to be a typo. I believe it should be "0.8%–4.2%." Please verify this.

- The term "algorithm ingesting data" is somewhat technical (my impression) for an abstract. To reach a broader audience, I suggest rephrasing it to "data was used as input to the algorithm" or something along these lines.

Author response:

Slight adjustments to the abstract were made to improve clarity and fix the typo.

Introduction

- "Relatively young sample of individuals" should be revised to "sample of relatively young individuals."

- Line 194: Correct the phrase to "daily diary."

Author response:

Both these textual adjustments have been made in the revised version of the manuscript.

Methodology

- Was a sample size analysis performed? While the number of people enrolled seems sufficiently large, there is a question about whether the study period was sufficient to observe a number of outcomes. I recommend including information on sample size analysis conducted.

Author response:

Section 4.4 of the appended protocol provides information on the sample size calculations. Traditional power calculations could not be performed due to the statistical methodology, and the fact that power was highly dependent on volatile factors (including the infection rate, which was unknown and constantly changing with new variants of the virus). We have therefore decided not to report on the sample size in the current manuscript.

- The description of "per-day analysis" is unclear. I understand it as calculating the number of infected days across all participants, which serves as a measure of person-time. Please verify if this interpretation is correct and consider clarifying the description in the text and in Table 5.

Author response:

This interpretation is correct. We have adjusted the text and added rows to Table 5 for additional clarifications.

- Another suggestion is to report the Youden index (sensitivity + specificity - 1) and indicate under which circumstances this index was positive (or exceeded a defined cutoff).

Author response:

We agree with the reviewer that methods such as the Youden index could have been used for improving the study algorithm and reporting on its comparative performance. However, no cutoff was defined during the study itself, and we currently are not able to further investigate algorithm performance with various cutoffs as the algorithm’s underlying code and probability outputs are proprietary. We have therefore chosen not to report the Youden index in the current paper, but have instead listed the use of this index as a suggestion for algorithm improvement in line 431.

Results

- Figure 2 requires higher image resolution or better text contrast. When I attempted to download it to try to see in better resolution, only the Figure 3 was downloaded. I double checked it but I am not sure why this happens.

Author response:

A new version of the figure has been uploaded with the revised manuscript.

- In S1 Appendix, I suggest adding to the captions that "survival probability" refers to the probability that the event has not occurred.

Author response:

We thank the reviewer for this suggestion and have added figure legends in the appendix accordingly.

Discussion

As indicated above, I would like to see a clearer outline of how to fine-tune the protocol, as well as metrics for a cost-effectiveness analysis.

Author response:

The discussion has been extended with an additional outline of methods for improving the algorithm (e.g., line 429). Cost-effectiveness analysis is beyond the scope of the current paper, but is mentioned in Line 420, with a reference to a (preprint) publication on the economic evaluation of the study.

Reviewer 2

Main strengths of the paper:

1) Innovative concept and relevance

The COVID-RED study embodies an innovative approach by leveraging wearable technology and artificial intelligence (AI) to address a critical public health issue: the early detection of SARS-CoV-2 infections. This research is particularly relevant in light of the COVID-19 pandemic, where the ability to identify infections during the pre-symptomatic or asymptomatic phase could significantly curb transmission rates. The study bridges advanced AI algorithms with wearable medical devices, showcasing a futuristic and interdisciplinary method for monitoring public health. This integration of technology with healthcare represents a paradigm shift from traditional diagnostics to more proactive and predictive health monitoring.

2) Robust study design

The study design is meticulously structured to ensure reliability and scientific rigor:

- Randomized Controlled Trial (RCT): A prospective, single-blinded, two-period, two-sequence, randomized crossover trial was employed. This robust design minimizes biases and allows a direct comparison between the experimental (wearable-based AI algorithm) and control (symptom-based) conditions.

- Large Sample Size: The inclusion of 17,825 participants is a remarkable achievement, providing high statistical power and the ability to generalize findings across populations.

- Crossover Nature: The two-period crossover design ensures that each participant serves as their control, reducing variability and enhancing the internal validity of the results.

- These features collectively underscore the study's scientific robustness and provide a solid foundation for its conclusions.

3) Use of cutting-edge wearable technology

The article highlights the use of the Ava bracelet, an FDA-cleared and CE-certified wearable device originally designed for fertility tracking, repurposed for infection detection. The bracelet's ability to measure multiple physiological parameters, including respiratory rate, heart rate, heart rate variability, wrist-skin temperature, and skin perfusion, is a testament to its advanced capabilities. The study emphasizes the bracelet’s potential for continuous health monitoring, enabling real-time alerts for possible infections. Repurposing existing technology for

---

## [Decision Letter · Decision Letter 1]

19 Mar 2025

PONE-D-24-38787R1A wearable-based AI algorithm for the remote early detection of SARS-CoV-2 infections: results from the COVID-RED study, a prospective randomised single-blinded crossover trialPLOS ONE

Dear Dr. Zwiers,

Thank you for submitting your manuscript to PLOS ONE. After careful consideration, we feel that it has merit but does not fully meet PLOS ONE’s publication criteria as it currently stands. Therefore, we invite you to submit a revised version of the manuscript that addresses the points raised during the review process.

We look forward to receiving your revised manuscript.

Kind regards,

Cecilia Acuti Martellucci, M.D.

Academic Editor

PLOS ONE

Additional Editor Comments:

I apologise for taking a long time to make a decision. I believe the manuscript was greatly improved upon revision, however I also agree with Reviewers 1 and 4. If this work is to contribute to the developement of the relative research domain, it should provide the details necessary for reproducibility.

Reviewers' comments:

Reviewer's Responses to Questions

**Comments to the Author**

1. If the authors have adequately addressed your comments raised in a previous round of review and you feel that this manuscript is now acceptable for publication, you may indicate that here to bypass the “Comments to the Author” section, enter your conflict of interest statement in the “Confidential to Editor” section, and submit your "Accept" recommendation.

Reviewer #1: (No Response)

Reviewer #2: All comments have been addressed

Reviewer #3: All comments have been addressed

Reviewer #4: (No Response)

2. Is the manuscript technically sound, and do the data support the conclusions?

Reviewer #1: No

Reviewer #2: Yes

Reviewer #3: Yes

Reviewer #4: Yes

3. Has the statistical analysis been performed appropriately and rigorously? 

Reviewer #1: Yes

Reviewer #2: Yes

Reviewer #3: Yes

Reviewer #4: No

4. Have the authors made all data underlying the findings in their manuscript fully available?

Reviewer #1: Yes

Reviewer #2: Yes

Reviewer #3: Yes

Reviewer #4: No

5. Is the manuscript presented in an intelligible fashion and written in standard English?

Reviewer #1: Yes

Reviewer #2: Yes

Reviewer #3: Yes

Reviewer #4: Yes

6. Review Comments to the Author

Reviewer #1: Several questions and concerns were raised in my review of the first submission regarding the methodology and the evaluation of the results in assessing the algorithm. I had hoped to see these questions addressed to better understand its performance and the reasons for the observed low specificity. This understanding is really important for a proper evaluation of the results. In fact, another review noted that the title of the manuscript specifically refers to the algorithm. Unfortunately, many of these issues cannot be fully addressed due to the proprietary nature of the algorithm, as stated in the authors’ response. Furthermore, additional evaluations are not possible for the same reason. I see this lack of description regarding the algorithm as a major weakness.

Reviewer #2: During the first review, I have already accepted the original submission with specific comments and do not have more remarks to add.

Reviewer #3: All my comments have been addressed by the authors and I recommend this paper to be accepted by PLOS One journal.

Reviewer #4: Dear authors,

Thank you for your responses. I appreciate your effort in addressing the raised concerns and making the necessary revisions. However, there still remains the issue regarding the reporting of the ROC AUC.

Research is built on the ability of others to understand and evaluate findings. Allowing for the reproducibility of your methodology to be disregarded due to the proprietary nature of your product, still either intentionally or unintentionally omitting standard performance metrics, especially one as widely used as AUC, compromises transparency. Even if the internal workings of an algorithm or the probability outputs are proprietary, reporting aggregate performance metrics does not reveal trade secrets.

Moreover, as you have elaborated in the introduction by reviewing other studies and comparing their reported AUCs, it is standard practice to report this aggregate metric. Future studies, as recommended in your discussion, would not have a sound basis to compare their model with the one developed here if this metric is omitted. This aggregate metric allows readers and researchers to fairly compare the proposed method to others. Not reporting this metric leaves a gap in the evaluation of the method's effectiveness. This issue directly affects the evaluation of your method's performance and the validity of the comparisons drawn thereafter. Addressing it is essential for upholding the manuscript's scientific rigor.

Finally, regarding the technical aspect of this issue, whatever environment is employed to develop an RNN can also be used to calculate AUC, as it provides “probability outputs” for the model’s predictions.

I should also remind you that the last line of methods (line 320) still lists AUC as a metric you were supposed to calculate and report.

Best regards

7. PLOS authors have the option to publish the peer review history of their article (what does this mean? ). If published, this will include your full peer review and any attached files.

**Do you want your identity to be public for this peer review?** For information about this choice, including consent withdrawal, please see our Privacy Policy .

Reviewer #1: No

Reviewer #2: **Yes: ** AXEL MOYAL

Reviewer #3: No

Reviewer #4: **Yes: ** Shahrokh Mousavi

---

## [Author Response · Author response to Decision Letter 2]

9 Apr 2025

Editor Comments

I apologise for taking a long time to make a decision. I believe the manuscript was greatly improved upon revision, however I also agree with Reviewers 1 and 4. If this work is to contribute to the development of the relative research domain, it should provide the details necessary for reproducibility.

Author response:

We thank the editor for their kind words on the improvements made during the first round of revisions. We provide responses to the comments from Reviewers 1 and 4 below. We have provided as much detail as possible for reproducibility and have made anonymised study data publicly available for future researchers. By doing this, we believe this works contributes to the scientific literature in the field.

Reviewer 1

Several questions and concerns were raised in my review of the first submission regarding the methodology and the evaluation of the results in assessing the algorithm. I had hoped to see these questions addressed to better understand its performance and the reasons for the observed low specificity. This understanding is really important for a proper evaluation of the results. In fact, another review noted that the title of the manuscript specifically refers to the algorithm. Unfortunately, many of these issues cannot be fully addressed due to the proprietary nature of the algorithm, as stated in the authors’ response. Furthermore, additional evaluations are not possible for the same reason. I see this lack of description regarding the algorithm as a major weakness.

Author response:

The reviewer correctly notes that the proprietary nature of the algorithm poses a limitation to our study. Despite this limitation, the authors still feel that the current manuscript adds to the scientific literature by providing such a large-scale evaluation of the use of a wearable device for SARS-CoV-2 infection detection. To provide more explanation for the low specificity of the algorithm, we have added some clarification about the chosen probability threshold in the development of the algorithm in line 245. In addition, we have added a statement about this to the discussion (line 423) as well. These additions hopefully better clarify the reasons for the high sensitivity and low specificity of the algorithm better, and these insights might benefit researchers working on similar projects in the future.

Regarding the title of the current manuscript, we understand that this might refer to the AI algorithm itself too strongly. However, we do feel it is worthwhile to mention it in the title as the wearable-based algorithm is the main intervention that is being evaluated in this trial and thus informs the reader accordingly. A potential alternative title could be “Remote early detection of SARS-CoV-2 infections using a wearable-based algorithm: results from the COVID-RED study, a prospective randomised single-blinded crossover trial”. If the editor has a preference for this alternative title, or has an alternative suggestion, we are open to discussing this modification.

Reviewer 2

During the first review, I have already accepted the original submission with specific comments and do not have more remarks to add.

Author response:

We thank the reviewer for their approval of the original and current submission.

Reviewer 3

All my comments have been addressed by the authors and I recommend this paper to be accepted by PLOS One journal.

Author response:

We thank the reviewer for this recommendation.

Reviewer 4

Dear authors,

Thank you for your responses. I appreciate your effort in addressing the raised concerns and making the necessary revisions. However, there still remains the issue regarding the reporting of the ROC AUC.

Research is built on the ability of others to understand and evaluate findings. Allowing for the reproducibility of your methodology to be disregarded due to the proprietary nature of your product, still either intentionally or unintentionally omitting standard performance metrics, especially one as widely used as AUC, compromises transparency. Even if the internal workings of an algorithm or the probability outputs are proprietary, reporting aggregate performance metrics does not reveal trade secrets.

Moreover, as you have elaborated in the introduction by reviewing other studies and comparing their reported AUCs, it is standard practice to report this aggregate metric. Future studies, as recommended in your discussion, would not have a sound basis to compare their model with the one developed here if this metric is omitted. This aggregate metric allows readers and researchers to fairly compare the proposed method to others. Not reporting this metric leaves a gap in the evaluation of the method's effectiveness. This issue directly affects the evaluation of your method's performance and the validity of the comparisons drawn thereafter. Addressing it is essential for upholding the manuscript's scientific rigor.

Finally, regarding the technical aspect of this issue, whatever environment is employed to develop an RNN can also be used to calculate AUC, as it provides “probability outputs” for the model’s predictions.

I should also remind you that the last line of methods (line 320) still lists AUC as a metric you were supposed to calculate and report.

Best regards

Author response:

We thank the reviewer for their suggestions and agree that reporting the AUC would be an improvement of the manuscript. As we indicated in our initial rebuttal, this performance metric was not initially considered for reporting due to its limited value for the evaluation of the interventions on top of the other performance metrics reported and the challenges faced in this study which compromised its interpretation. For example, the decision thresholds of the algorithms could not easily be informed by the AUC as there was also a necessity for the wearable-based AI algorithm to be sensitive enough to catch (asymptomatic) infections on top of what would normally be reported through symptom evaluation. In addition, given the fundamental differences between the control and experimental condition, the AUC had little added value in the comparison or evaluation of these algorithms. Nevertheless we agree that sharing the AUC for transparency and reproducibility reasons would be an improvement of the manuscript. However, while it is indeed the case that sharing the AUC would not reveal trade secrets, the company that owns the algorithm code (Ava AG) has been acquired by another company and all individuals who have contributed to the current study have subsequently left that company. This has prevented us from even discussing the possibility of sharing the model probability outputs, which were not shared during the study as these were considered proprietary. We thus unfortunately do not have access to the probability outputs needed to calculate the AUC. The authors have investigated if there were any ways to still report the AUC, but are unable to do so at this moment, unfortunately. We are aware that this is a limitation to our manuscript, as it leaves a gap in the algorithm evaluation. We have made some adjustments to the manuscript to address this, which are explained below.

Firstly, the reviewer noted that the methods still listed AUC as a metric of interest, which was incorrect. We have removed this metric from the list in line 325 of the manuscript.

Additionally, we have further addressed the limitation of not reporting the AUC in the discussion of our manuscript by acknowledging that we cannot fully compare the performance of this study’s algorithm to that of previous literature (line 442). However, given the extremely low specificity, it can still be concluded that the study algorithm does not perform well enough to be used in practice, even without calculating the AUC.

---

## [Decision Letter · Decision Letter 2]

8 May 2025

Remote early detection of SARS-CoV-2 infections using a wearable-based algorithm: results from the COVID-RED study, a prospective randomised single-blinded crossover trial

PONE-D-24-38787R2

Dear Dr. Zwiers,

We’re pleased to inform you that your manuscript has been judged scientifically suitable for publication and will be formally accepted for publication once it meets all outstanding technical requirements.

Kind regards,

Cecilia Acuti Martellucci, M.D.

Academic Editor

PLOS ONE

Additional Editor Comments (optional):

Reviewers' comments:

Reviewer's Responses to Questions

**Comments to the Author**

1. If the authors have adequately addressed your comments raised in a previous round of review and you feel that this manuscript is now acceptable for publication, you may indicate that here to bypass the “Comments to the Author” section, enter your conflict of interest statement in the “Confidential to Editor” section, and submit your "Accept" recommendation.

Reviewer #1: All comments have been addressed

Reviewer #4: All comments have been addressed

2. Is the manuscript technically sound, and do the data support the conclusions?

Reviewer #1: Yes

Reviewer #4: Yes

3. Has the statistical analysis been performed appropriately and rigorously? 

Reviewer #1: Yes

Reviewer #4: No

4. Have the authors made all data underlying the findings in their manuscript fully available?

Reviewer #1: Yes

Reviewer #4: No

5. Is the manuscript presented in an intelligible fashion and written in standard English?

Reviewer #1: Yes

Reviewer #4: Yes

6. Review Comments to the Author

Reviewer #1: In my opinion, the title in the revised manuscript is better.

Reviewer #4: Dear Authors,

Thank you for your detailed responses and for the recent revisions made to your manuscript. While I remain somewhat concerned about the omission of the ROC AUC metric, I understand the constraints imposed by the proprietary nature of the model outputs and the associated practical challenges.

Despite these concerns, I find that the revisions you have implemented sufficiently address the issues raised.

Best regards

7. PLOS authors have the option to publish the peer review history of their article (what does this mean? ). If published, this will include your full peer review and any attached files.

**Do you want your identity to be public for this peer review?** For information about this choice, including consent withdrawal, please see our Privacy Policy .

Reviewer #1: No

Reviewer #4: **Yes: ** Shahrokh Mousavi

---

## [Editor Report · Acceptance letter]

PONE-D-24-38787R2

PLOS ONE

Dear Dr. Zwiers,

I'm pleased to inform you that your manuscript has been deemed suitable for publication in PLOS ONE. Congratulations! Your manuscript is now being handed over to our production team.

Kind regards,

on behalf of

Dr. Cecilia Acuti Martellucci

Academic Editor

PLOS ONE